# Risk Assessment of Toxic Heavy Metal Exposure in Selected Seafood Species from Thailand

**DOI:** 10.3390/foods14213725

**Published:** 2025-10-30

**Authors:** Alongkote Singhato, Narisa Rueangsri, Purimprat Thanaratsotornkun, Konpong Boonyingsathit, Piyanut Sridonpai, Nunnapus Laitip, Nattikarn Ornthai, Kunchit Judprasong

**Affiliations:** 1Department of Nutrition and Dietetics, Faculty of Allied Health Sciences, Burapha University, Mueang, Chonburi 20131, Thailand; alongkote@go.buu.ac.th (A.S.); narisar@go.buu.ac.th (N.R.); 2Institute of Nutrition, Mahidol University, Phutthamonthon, Nakhon Pathom 73170, Thailand; purimprat.nut@gmail.com (P.T.); konpong.boo@mahidol.ac.th (K.B.); piyanut.sri@mahidol.ac.th (P.S.); 3Chemical Metrology and Biometry Department, National Institute of Metrology (Thailand), Klong Luang, Pathum Thani 12120, Thailand; nunnapusl@nimt.or.th (N.L.); nattikarn@nimt.or.th (N.O.)

**Keywords:** heavy metals, seafood contamination, risk assessment, good health

## Abstract

This study evaluates the risk of toxic heavy metal exposure in 20 commonly consumed seafood species from Thailand, focusing on arsenic (As), cadmium (Cd), mercury (Hg), and lead (Pb). Seafood is nutritionally valuable but may accumulate harmful metals due to environmental contamination from industrial, agricultural, and medical sources. Samples were collected from markets in Chonburi, prepared through boiling, frying, and grilling, and analyzed using ICP-MS/MS. Most toxic metal levels were within Thai regulatory limits; however, Wedge shell and Musk crab showed arsenic concentrations exceeding permissible levels. Risk assessment employed hazard quotient and margin of exposure calculations using consumption data stratified by age and cooking methods. Results demonstrated that arsenic presents the highest risk, particularly for children aged 0–5.9 years, with Wedge shell and Musk crab posing significant concerns. Cadmium and mercury generally posed low or no risk across samples, except for isolated high-level cadmium exposure in Wedge shell and occasional mercury concerns. Lead, based on the margin of exposure assessment, showed significant health risk for eater only group. The study concludes that although arsenic contamination in certain seafood species requires careful monitoring and public awareness, other toxic metals—particularly cadmium, mercury, and lead—currently present minimal health risks. Routine surveillance of seafood contaminants is essential to safeguard consumer health, particularly among vulnerable groups.

## 1. Introduction

Marine-derived foods have played a significant role in the traditional Thai diet for over 3000 years, as indicated by archeological findings such as the fossilized remains of striped-head fish species discovered in ancient settlements [1]. As a dietary group, marine fish and other seafood species are recognized as rich sources of high-quality proteins, essential minerals (e.g., iodine, selenium, and zinc), and a wide range of vitamins, including vitamin D and B-complex vitamins [2]. Moreover, they provide biologically important long-chain omega-3 polyunsaturated fatty acids, particularly docosahexaenoic acid (DHA) and eicosapentaenoic acid (EPA), which have been extensively studied for their cardioprotective properties [3]. These fatty acids contribute to the regulation of blood lipid profiles, reduction in systemic inflammation, and maintenance of endothelial function, thereby lowering the risk of cardiovascular diseases. Given their comprehensive nutritional profile and health-promoting benefits, the regular consumption of marine fish and seafood is strongly advocated in public health guidelines to support the nutritional well-being of individuals across all age groups.

Significant industrial and infrastructure development, as well as the encouragement of export-oriented companies, have been driven by Thailand’s economic progress. However, insufficient strategic planning has resulted in the degradation of water resources and aquatic ecosystems, primarily because the release of toxic substances from industrial activities (e.g., plastics, machinery, paints, PVC, and batteries), agricultural inputs (e.g., insecticides and fertilizers), and products from the medical and healthcare sectors (e.g., pharmaceuticals, medical equipment, and cosmetics). Among various environmental contaminants, arsenic (As), cadmium (Cd), lead (Pb), and mercury (Hg) are considered particularly hazardous with respect to food safety regulations. These heavy metals are especially detrimental to the human nervous and urinary systems [4,5,6,7]. Chronic ingestion of low concentrations of inorganic As has been associated with dermatological manifestations (e.g., hyperpigmentation, hyperkeratosis, corns, and warts) and peripheral neuropathy, which may initially present as numbness in the extremities and later progress to a painful paresthesia [5]. Cadmium exposure is linked to renal tubular and glomerular damage, impaired bone mineralization, elevated fracture risk, compromised pulmonary function, and emphysema, typically following prolonged exposure. Exposure to lead has been associated with a wide range of adverse health effects, including impaired cognitive function, alterations in mood and behavior, disturbances in neuromotor and neurosensory activities, decreased glomerular filtration rate, elevated blood pressure, inhibition of key enzymes involved in heme synthesis, diminished sperm production, and an increased likelihood of spontaneous abortion [6]. Mercury exerts toxic effects on the nervous, digestive, and urinary systems. In response to these health risks, Thailand’s Ministry of Public Health has established maximum allowable concentrations for As, Cd, Pb, and Hg in fish and seafood at 2.0, 1.0, 0.3, and 0.5 mg/kg, respectively, in accordance with official regulatory standards, which are consistent with the guidelines recommended by the World Health Organization (WHO) [8].

Many studies have examined the presence of harmful substances in Thailand’s regularly eaten seafood and marine fish species in light of the escalating public health concerns. For instance, an examination of fish samples taken from Bangkok’s commercial markets revealed that the concentrations of As in Nile tilapia, red tilapia, striped snakehead, king mackerel, sea bass, and grouper were, respectively, 0.50, 1.76, 0.31, 3.11, 11.65, and 4.42 mg/kg. The comparable values of Hg were 0.90, 1.13, 0.75, 1.37, <0.50, <0.50, and 0.90 mg/kg, respectively [9]. Notably, striped snakehead, king mackerel, sea bass, and grouper had Hg levels above the legal threshold of 0.5 mg/kg, whereas As levels in these fish above the maximum allowable limit of 2 mg/kg [9].

The contamination of seafood with toxic elements is affected by multiple variations, including the geographical location of aquatic habitats, the level of environmental pollution at capture sites, prevailing ecological conditions, species-specific biological characteristics, and even household cooking practices. Currently, there is a limited number of studies focusing exclusively on toxic heavy metals and associated exposure risks in fish consumed in Thailand [10]. However, other commonly consumed seafood species have not been thoroughly investigated. To address this gap and generate data a more comprehensive evaluation of potential health risks to people, this study analyzed the concentrations of As, Cd, Hg, and Pb in additional commonly consumed seafood species using inductively coupled plasma triple quadrupole mass spectrometry (ICP-MS/MS). These elements are commonly found as contaminants in food and have received considerable attention in previously mentioned studies. Furthermore, the study evaluated the associated exposure risks for the Thai population resulting from the consumption of these seafood species.

## 2. Materials and Methods

### 2.1. Seafood Sample Purchasing and Preparation

Twenty species of seafood commonly farmed, captured, and widely distributed in local markets throughout Thailand were selected for this study, based on national aquaculture production statistics [11,12] and the Thai Food Composition Database [13]. Table 1 provides the scientific, common, and Thai names of each selected species. Representative samples (500 g per purchase) were randomly collected from three to four vendors at three seafood markets located in Chonburi Province, an eastern coastal region of Thailand. Sample preparation and cooking procedures were conducted in accordance with established protocols previously described by related studies [14], and were performed at the Department of Nutrition and Dietetics, Burapha University.

Briefly, each seafood specimen was weighed (in grams) before and after the removal of non-edible portions. For thermal processing, boiling was conducted in deionized water (Milli-Q^®^ EQ 7000, Merck KGaA, Darmstadt, Germany), frying was performed using palm oil, and grilling was carried out on an electric grill pan. Following cooking, the samples were homogenized with a high-speed homogenizer (Ultra-Turrax T25, IKA, Staufen, Germany) and subsequently freeze-dried (Kinetic Engineering Co., Ltd., Bangkok, Thailand) with controlled parameters at −55 °C and 0.040 mbar for 24 h. The resulting freeze-dried materials were finely milled using a laboratory grinder (A11 Basic Analytical Mill, IKA, Staufen, Germany), placed in screw-capped plastic containers, and kept at −20 °C until analysis.

### 2.2. Chemicals and Reagents

High-purity nitric acid (Suprapure, 65% HNO_3_) used for sample digestion was supplied by Sigma-Aldrich (St. Louis, MO, USA). Standard stock solutions of As, Cd, Hg, Pb, and rhodium (Rh)—the latter serving as an internal standard to ensure analytical reliability—were obtained from the National Institute of Standards and Technology (NIST, Gaithersburg, MD, USA). To assess the accuracy and precision of the analytical procedure, certified reference material (CRM) SRM 1566b (oyster tissue) from NIST was analyzed. Additionally, CRM NMIJ 7402-a (codfish tissue) acquired from the National Metrology Institute of Japan (NMIJ) was employed for quality assurance in seafood matrix verification. Ultrapure water (Milli-Q^®^, MilliporeSigma, Billerica, MA, USA) was consistently used throughout the experimental process for sample preparation and dilution to minimize contamination and maintain analytical integrity.

### 2.3. Determination of Moisture

Determination of moisture content obtained in samples was performed following the standard gravimetric method. Initially, samples were pre-dried using sand in a water bath, followed by further drying in a hot-air oven (Memmert ULE 400, Memmert GmBhH, Büchenbach, Germany) at 100 ± 2 °C for 2 h. After drying, the samples were cooled in a desiccator and subsequently weighed using a four-digit analytical balance (Mettler AT201, Mettler Toledo, Hamilton, New Zealand). The drying and weighing process was repeated at one-hour intervals until a constant weight was achieved, in accordance with AOAC Official Method 925.23 (AOAC, 2023) [15].

### 2.4. Analysis of Toxic Element Levels

The contents of total As, Cd, Hg, and Pb in the selected seafoods were determined based on the AOAC Official Method 2015.01 (AOAC, 2023) [15]. Approximately 0.25 g of each sample was transferred into microwave digestion vessels. To each vessel, 4 mL of concentrated HNO_3_ and 1 mL of 30% hydrogen peroxide (H_2_O_2_) were added, followed by the addition of 0.1 mL of an internal standard solution containing rhodium (Rh) at a concentration of 50 mg/L in 5% (*v*/*v*) HNO_3_. The vessels were then securely sealed and subjected to microwave-assisted digestion, with under condition described elsewhere [16]. After digestion, the vessels were allowed to cool to ambient temperature before being carefully opened. The resulting digests were quantitatively transferred into acid-cleaned 50 mL high-density polyethylene (HDPE) centrifuge tubes and subsequently diluted with deionized water to a final volume of 20 mL. Determination of toxic element concentrations was performed using inductively coupled plasma triple quadrupole mass spectrometry (ICP-MS/MS; Agilent 8800 ICP-QQQ, Agilent Technologies, Waldbronn, Germany). The operating conditions was presented in the previous study [16] and analytical results were reported as milligrams per kilogram (mg/kg) on a fresh weight (FW) basis [16].

To validate the analytical method, toxic element recoveries from spiked samples were assessed. The recovery rates for As, Cd, Hg, and Pb were 90 ± 5% (range: 85–95%), 100 ± 5% (95–105%), 111 ± 4% (107–115%), and 110 ± 4% (106–113%), respectively, all within the respectable range of 80–115%. The method detection limits (LOD), defined as three times the standard deviation (3SD) of ten replicate measurements at the lowest detectable concentration, were 0.0007, 0.0009, 0.0011, and 0.0011 mg/kg FW for As, Cd, Hg, and Pb, respectively. The limits of quantitation (LOQ), defined as ten times the standard deviation (10SD) of ten replicate measurements at the lowest detectable concentration, were 0.0063, 0.0062, 0.0124, and 0.0123 mg/kg FW for As, Cd, Hg, and Pb, respectively.

### 2.5. Risk Assessment of the Toxic Heavy Metals Exposure

The risk evaluation was conducted by equating the estimated total exposure of each toxic element with its corresponding health-based guidance value (*HBGV*) established by the Joint FAO/WHO Expert Committee on Food Additives (JECFA) [17]. The overall dietary exposure to individual toxic elements through seafood consumption was determined following the procedure described in reference [18], as expressed in Equation (1).(1)$$TE=\frac{C\times DI}{BW}$$

In this context, *TE* represents the total exposure to a toxic element (mg/kg body weight), *C* denotes concentration of the toxic metal element in the seafood (mg/kg). In the risk assessment, the wet weight concentration is used, *DI* is the daily consumption of food (kg/day), and *BW* refers to body weight (kg).

The health risk associated with exposure to personal heavy metals through seafood intake was further evaluated using the hazard quotient (*HQ*), calculated as a percentage of the health-based guidance value (*HBGV*), according to the method described in reference [18] (Equation (2)). An *HQ* value exceeding 1.0 was interpreted as an indication that the level of the toxic element posed a potential health risk [19], rendering the food unsafe for regular consumption.(2)$$HQ=\frac{TE\times P}{HBGV}$$

In this context, *HBGV* refers to the health-based guidance value (µg/kg body weight), which may be defined as a tolerable daily consumption, tolerable weekly consumption, or tolerable monthly consumption, depending on the heavy metals in question. The variable *P* represents the relevant exposure period: 1 for daily, 7 for weekly, and 30 for monthly assessments. Toxicological studies have identified the primary target organ of cadmium as the renal proximal tubule. Chronic exposure to cadmium causes renal tubular dysfunction. The joint FAO/WHO Expert Committee on Food Additives (JECFA) has established the Health-Based Guidance Value (*HBGV*) as a Tolerable Monthly Intake (*PTMI*) of 25 µg/kg body weight per month to prevent adverse effects on kidney function from chronic cadmium exposure [20], while JECFA (2011) [20] evaluated inorganic mercury based on toxicological studies of the renal proximal tubules in humans and experimental animals, and established a Provisional Tolerable Weekly Intake (*PTWI*) of 4 µg/kg body weight per week to prevent chronic renal toxicity from dietary exposure to inorganic mercury [21]. In contrast, the *HBGV*s for Pb and As have been withdrawn due to emerging toxicological concerns and insufficient data to establish safe thresholds [17].

As a result, the previously described equation could not be applied to As and Pb; instead, the margin of exposure (*MOE*) was employed as an optional risk assessment metric. The *MOE* was calculated in accordance with the method described in reference [17], as shown in Equation (3). For lead, an *MOE* value below 100 suggests a potential or actual health risk associated with its consumption, whereas an *MOE* value exceeding 100 indicates a low level of concern for public health [22]. Regarding arsenic, EFSA’s assessment defined health concern thresholds with *MOE* values ranging from 2.0 to 0.4 for average consumers and from 0.9 to 0.2 for high consumers, indicating potential health risks that require careful monitoring [23].(3)$$MOE=\frac{BMDL}{TE}$$

In this context, *MOE* represents the margin of exposure, and *BMDL* denotes the benchmark dose lower limit for inorganic As, set at 0.06 µg/kg body weight per day. This value is derived from human skin cancer studies and is used as a Reference Point (RP) for risk assessment. Continuous exposure to inorganic arsenic through diet is associated with an increased risk of skin cancer, bladder cancer, and lung cancer [22].

The Margin of Exposure (*MOE*) for lead was assessed using Benchmark Dose Lower Confidence Limits (*BMDL*s) of 0.5 µg/kg body weight per day to evaluate developmental neurotoxicity, applicable for children aged 3 to 17.9 years in this study. Additionally, a *BMDL* of 0.63 µg/kg body weight per day was employed to assess chronic kidney disease risk in adult consumers aged 18 years and above. These *BMDL*s provide toxicological reference points to support health risk assessments in the studied population [24].

In the assessment of consumer health risk, the evaluation should be based on the levels of contaminants in foods as consumed, i.e., in the cooked form. Since this study analyzed heavy metal concentrations in raw samples, the results were adjusted using cooking yield factors to account for changes in metal concentrations after cooking by boiling, frying, and grilling. The applied factors were obtained from previous studies as presented in Table 2 of the referenced document [16] and in Appendix A.

The summary of *HBGV*s for the heavy metals studied is presented in Table 2. This table presents for each heavy metal, which are standards established by food safety and health organizations. These values serve as guidelines for assessing health risks from exposure to heavy metals in food or the environment. The specific standards vary depending on the type of heavy metal. These values reflect levels of consumption or exposure to heavy metals that are considered acceptable without causing long-term adverse health effects. Exceeding these standards may increase the risk of toxicological effects, such as disorders of the nervous system, kidneys, liver, and reproductive system.

The seafood consumption data used in this study were obtained from the National Bureau of Agricultural Commodity and Food Standards (ACFS), the official agency responsible for collecting and publishing Thai population food consumption statistics [12]. The consumption data are divided into two types: per capita consumption, representing the average seafood intake of the entire population regardless of regular consumption status (shown in Table 3); and eater-only consumption, reflecting the actual intake of the subgroup that consumes seafood (shown in Table 4). Additionally, average body weight data for each age group, an important factor for exposure assessment, are presented in Table 5. These data support the health risk assessment of heavy metal contamination through seafood consumption.

### 2.6. Statistical Analyses

To ensure precision and reproducibility, every analytical measurement was carried out in triplicate. The mean values ± standard deviation (SD) were used to report the levels of hazardous components in samples taken from the three marketplaces. To identify significant differences at a confidence level of *p* < 0.05, one-way analysis of variance (ANOVA) was used to compare the mean concentrations for each seafood species. This was followed by Duncan’s multiple range test. SPSS software, version 24.0 for Windows (SPSS Inc., Chicago, IL, USA), was used for all statistical analyses.

## 3. Results

### 3.1. Moisture and Toxic Heavy Element Concentration

Depending on the kind of seafood, the moisture content of the 20 species ranged from 64.1% to 88.0% (Table 6). Table 6 also displays the levels of total As, Cd, Hg, and Pb found in the seafood samples.

The As concentrations in the selected seafood species ranged from 0.2 to 21.4 mg/kg fresh weight (FW). The three species with the highest As levels were wedge shell (21.4 ± 2.0 mg/kg FW), musk crab (16.2 ± 1.8 mg/kg FW), and Indo-Pacific horseshoe crab (eggs) (10.6 ± 1.7 mg/kg FW).

Regarding Cd content, all seafood species showed detectable levels. The three species with the highest Cd concentrations were wedge shell (7.3 ± 1.8 mg/kg FW), cockle (1.7 ± 0.3 mg/kg FW), and clam (1.3 ± 0.6 mg/kg FW).

The Hg concentrations were quantifiable in most seafood species, except for Pacific white shrimp, banana prawn, giant tiger prawn, and serrated mud crab, which had levels below the LOD and LOQ. Overall, Hg concentrations in the analyzed species ranged from below the LOQ to a maximum of 0.1 mg/kg FW.

Regarding Pb concentrations, all seafood species exhibited quantifiable levels of approximately 0.1 mg/kg FW, except for splendid squid and cockle, which had higher levels at 0.2 mg/kg FW.

### 3.2. Risk Assessment of Toxic Elements Through Seafood Consumption

#### 3.2.1. Comparison of Toxic Element Content with Maximum Limit

The analysis of selected seafood samples revealed that the majority of heavy metal concentrations were below the standards stipulated in Thai Ministry of Public Health Notification No. 414 [8]. This regulation sets maximum permissible levels for contaminants, including inorganic arsenic (2.0 mg/kg), cadmium in bivalves and squid (2.0 mg/kg), cadmium in fish (1.0 mg/kg), mercury (0.5 mg/kg), and lead (1.0 mg/kg, with a lower threshold of 0.3 mg/kg for fish). The measured concentrations were as follows: inorganic arsenic (0.01–1.1 mg/kg FW), cadmium (0.001–7.39 mg/kg FW), mercury (0.001–0.06 mg/kg FW), and lead (0.01–0.22 mg/kg FW). While most samples complied with these standards, the cadmium concentration in the Wedge shell sample (7.39 mg/kg) exceeded the regulatory limit of 2.0 mg/kg for bivalves.

#### 3.2.2. Risk Assessment of As

The risk assessment of arsenic in seafood was conducted based on inorganic arsenic concentrations, calculated as a percentage of total arsenic: 5% for shrimp and prawns, crabs, squids, and shellfish groups, and 2% for marine fish, as referenced [25]. The risk characterization employed the Margin of Exposure (*MOE*) approach, using a *BMDL* value established by EFSA of 0.06 (µg/kg BW/day). EFSA’s assessment defined health concern thresholds whereby *MOE* values ranged from 2.0 to 0.4 for average consumers and from 0.9 to 0.2 for high consumers, indicating potential health risks that warrant careful monitoring [23]. The margin of exposure (*MOE*) values in Table 7 indicate varying levels of arsenic risk from different seafood species on a population basis. Appendix A presents *MOE* specifically for consumers of each seafood type, highlighting potential health risks for frequent eaters. These results support focused risk assessments and targeted food safety measures for arsenic in seafood.

The risk assessment used consumption data from the National Bureau of Agricultural Commodity and Food Standards, incorporating both per capita and eater-only consumption data. The inorganic arsenic risk assessment results for the study samples are shown in Table 7.

After assessing using per capita consumption data, among the Shrimp and Prawn group, the species with the highest risk was the Ornate Rock Lobster, which posed risks across all cooking methods and age groups except for the 65 years and older group consuming boiled/grilled, and the 35 years and older group consuming fried. For Pacific White Shrimp and Banana Prawn, risk was observed at average consumption levels for boiled preparation (ages 3–5.9 years), and at the 97.5th percentile for boiled (3–17.9 years) and fried/grilled (3–12.9 years). Giant Tiger Prawn showed no notable risk at average consumption, while risk was present at the 97.5th percentile for all cooking methods (3–12.9 years). In the Crabs group, the Musk Crab exhibited the highest risk, with average consumption showing risk from boiled/grilled (3–12.9 years) and fried (3–34.9 years), while the 97.5th percentile showed risk across all cooking methods and age groups. Blue Crab presented no significant risk at average consumption but did show risk at the 97.5th percentile for all cooking methods (3–12.9 years). Serrated Mud Crab and Red Frog Crab showed no concerning risk. For Squids, Bigfin Reef Squid had the highest risk, with average consumption risk across all cooking methods (3–34.9 years) and the 97.5th percentile showing risk across all age groups except boiled preparation for those aged 65 and older. Splendid Squid showed risk at average consumption for boiled (3–12.9 years), fried (3–17.9 years), and grilled (3–5.9 years), with the 97.5th percentile showing risk for boiled/grilled (3–34.9 years) and fried (3–64.9 years). Cuttlefish showed risk at average consumption for boiled/grilled (3–5.9 years) and fried (3–12.9 years), and the 97.5th percentile showed risk for all cooking methods (3–34.9 years). In the Shellfish group, risk was found for Cockle at the 97.5th percentile for boiled (6–12.9 years), fried (3–17.9 years), and grilled (3–34.9 years), and for Mussels at the 97.5th percentile for fried/grilled (3–5.9 years). Other shellfish species did not show concerning risk. The risk assessment for Marine Fish species, including Northern Whiting Fish and Silver Pomfret, indicated no significant health concerns for consumers.

Using eater-only consumption data, the risk assessment results indicated a generally high risk across all studied samples. However, some samples showed safety for consumers within the eater-only group, including Clam and Silver Pomfret. For Clam, both average consumption and the 97.5th percentile demonstrated safety when boiled or grilled for individuals aged 6 years and older. In the case of Silver Pomfret, safety was observed for boiling across all age groups, while fried preparation was safe for those aged 13 years and older, and grilled preparation was safe for those aged 6 years and older.

#### 3.2.3. Risk Assessment of Cd

This subsection frames risk characterization using both per-capita (consumers + non-consumers) and eater-only (consumers only) exposure metrics, each evaluated at the mean and 97.5th percentile consumption levels. Risk interpretation was based on Hazard Quotient (*HQ*) values, calculated relative to the health-based guidance value (*HBGV*) for cadmium, i.e., a provisional tolerable monthly intake (*PTMI*) of 25 µg/kg body weight/month. Within this framework, *HQ* ≥ 1 was considered to indicate potential health risk under the assessed consumption scenario, whereas *HQ* < 1 was interpreted as acceptable. Here, “mean” and “97.5th percentile” explicitly refer to statistics of consumption data rather than contaminant concentrations. Appendix A present the hazard quotient (*HQ*) of cadmium from each seafood species on a per capita basis and per eater-only basis, respectively. These tables highlight the potential health risks from cadmium exposure through seafood consumption for the general population and specific consumers. The data support risk assessments tailored to both population-wide and individual seafood consumer exposures.

Under the per capita framework, no seafood item exceeded *HQ* = 1 at mean consumption. At the 97.5th percentile (high consumption) level, however, discrete hotspots emerged, most prominently Cockle, which exceeded unity across all assessed cooking methods: boiled (risk in ages 6–34.9 y), fried (3–34.9 y), and grilled (3–64.9 y). Additional, method- and age-specific exceedances were observed for selected items, including Musk crab (fried/grilled, 3–5.9 y), Cuttlefish (fried, 3–5.9 y), and Ornate rock lobster (grilled, 3–12.9 y). Outside these hotspots, most per capita *HQ*s at the 97.5th percentile remained below 1 across cooking methods and age groups.

When restricted to the eater-only basis, consumption levels and the resulting exposures were higher than per capita values, yielding more frequent and larger exceedances at both the mean and, more markedly, the 97.5th percentile. Items showing consistent exceedances across multiple cooking methods and age strata included Musk crab, Oysters, Cockle, Wedge shell, and Red frog crab. Additional mean level exceedances were noted in specific age groups for Blue crab, Cuttlefish, Mussels, and Serrated mud crab. Several items exhibited method-specific sensitivity, for example, Ornate rock lobster (notably when fried/grilled), Indo Pacific horseshoe crab (eggs) (notably when fried), and Razor clam (notably at the 97.5th percentile). In aggregate, population average risk appears low, but high consumption scenarios and/or the eater-only perspective reveal nontrivial cadmium risk, particularly for young children, underscoring the importance of interpreting results within the statistical framing of the underlying consumption data.

#### 3.2.4. Risk Assessment of Hg

The risk assessment of mercury using the Provisional Tolerable Weekly Intake (PTWI), which in this case is 4 µg/kg body weight per week, indicates the maximum amount of mercury that consumers can be exposed to without causing long-term health effects [24,26]. This value is established based on toxicological data analysis and accumulation of the substance in the body. For quantitative risk assessment, the Hazard Quotient (*HQ*) is calculated; if *HQ* > 1, it indicates a health risk from exposure to the toxic substance because the exposure exceeds the safe level. Appendix A show the hazard quotient (*HQ*) for mercury exposure from different seafood species on a per capita and per eater-only basis, respectively. These tables illustrate the potential health risks posed by mercury through seafood consumption, providing a basis for both population-level and consumer-specific risk evaluation without overlapping content from other sections.

The mercury risk assessment results in the studied samples showed that when using per capita consumption data, there was no significant health risk for consumers at both average consumption levels and the 97.5th percentile consumption. However, when using eater-only consumption data, most samples also posed no concern, except for grilled cockle, which at the 97.5th percentile consumption level (ages 3–12.9 years) indicated a potential risk.

#### 3.2.5. Risk Assessment of Pb

Lead risk was evaluated using EFSA’s Margin of Exposure framework. *MOE* was defined as the endpoint-specific *BMDL* divided by dietary exposure to lead. This study applied a *BMDL* of 0.50 µg/kg bw/day for developmental neurotoxicity in consumers aged 3 to 17.9 years, and 0.63 µg/kg bw/day for chronic kidney disease in consumers aged 18 years and older. In this report, values above 100 indicate lower concern and values below 100 indicate potential concern.

Exposure was summarized across age strata using two statistics, the mean and the 97.5th percentile, with estimates reported for both per-capita and eater-only populations. Under the simplified two-tier rule, an entry is classified as low risk only if both statistics exceed 100 in every age group; all other entries are classified as at risk and warrant surveillance. Appendix A present the margin of exposure (*MOE*) for lead from various seafood species on a per capita and per eater-only basis, respectively. These tables provide insights into the potential lead exposure risks for the general population and specific seafood consumers, supporting focused risk evaluations for dietary lead intake.

In the per-capita view, cooking method differentiated risk: boiled showed the largest share of values above 100, grilled was intermediate, and fried the smallest. Applying the two-tier rule yielded a small set of low-risk exemplars: boiled, Shellfish (captured: Razor clam, Clam, Wedge shell) and marine fish (farmed: Silver pomfret); grilled, Shellfish (captured: Razor clam, Clam, Wedge shell); fried, Shellfish (captured: Razor clam, Wedge shell). All other per-capita combinations were classified as at risk.

In the eater-only view, conditioning on consumers produced lower values overall, with both statistics generally below 100 across many groups and age strata. A single age-specific exception exceeded 100: Marine fish (farmed: Northern whiting), boiled, in consumers aged 65 years and older (mean = 102.09). This narrow margin does not alter the overall conclusion that eater-only exposures commonly fall within ranges of potential concern across methods and ages. Consistent with the two-tier rule, no eater-only species-by-method combination met the low-risk threshold.

## 4. Discussion

### 4.1. Moisture and Toxic Element Contents

All seafood samples in the present study contained measurable levels of As, Cd, and Pb. Previous study has indicated that the concentrations of these toxic heavy metals are generally higher in marine organisms compared with freshwater species [9]. This difference may be attributed to the elevated availability of ionic sources of toxic elements in marine environments, which can promote their release from sediments into the aquatic ecosystem [13]. Ocean salinity influences the uptake of toxic elements by marine organisms [14]. Among the seafood species examined, the Wedge shell exhibited the highest concentrations of arsenic and cadmium, likely associated with its feeding habits, which include detritus, phytoplankton, and zooplankton [27]. Such feeding behaviors may contribute to biomagnification, the progressive accumulation of toxic elements along the food chain [28]. Furthermore, the Wedge shell’s habitat—burrowed within sandy or sandy–mud flats—may increase its exposure to sediment-associated toxic elements [29]. In contrast, clams and mussels typically inhabit fine sand or sandy–mud substrates in shallow waters and have distinct feeding strategies, which are associated with comparatively lower levels of biomagnification [30]. Musk crab, which inhabits mud flats, also exhibited elevated arsenic levels. This observation is consistent with previous studies reporting that swimming crabs, shrimp, and squid from a eutrophic Brazilian estuary were contaminated with toxic metals and metalloids, with crabs identified as the primary bioaccumulators [31]. For mercury, the higher levels of Hg in the analyzed seafood species can be primarily attributed to their trophic level, feeding habits, and habitat characteristics. Predatory species that feed on smaller fish tend to bioaccumulate methylmercury, resulting in higher concentrations. Moreover, species living in coastal or estuarine areas affected by anthropogenic pollution may also show elevated Hg levels. These factors, combined with the natural ability of mercury to bioaccumulate and biomagnify through the food chain [27,28,29,30,31].

### 4.2. Comparing the Legal Standard with the Toxic Element in Seafood

As previously indicated, the total arsenic concentrations in Wedge shell and Musk crab exceeded the Thai regulatory limit of 2 mg/kg [8]; however, this limit applies specifically to inorganic arsenic rather than total arsenic. Accordingly, the present study estimated the toxic element as inorganic arsenic (iAs), assuming it constitutes 10% of the total arsenic content in seafood [22]. Risk assessment of arsenic in seafood was performed based on these estimated inorganic arsenic concentrations, expressed as a proportion of total arsenic: 5% for shrimp, prawns, crabs, squids, and shellfish, and 2% for marine fish, as referenced [25]. Wedge shell and Musk crab presented the highest risk, with average consumption posing potential risks from boiled or grilled preparations (3–12.9 years) and fried preparations (3–34.9 years), whereas the 97.5th percentile of consumers faced risk across all cooking methods and age groups. Chronic exposure to elevated arsenic levels may accumulate in the human body and contribute to carcinogenesis [32,33,34].

### 4.3. Risk Evaluation of Toxic Substances via Seafood Intake

In the present seafood risk assessment, Wedge shell and Musk crab were identified as posing a considerable arsenic-related health concern for consumers across all age groups. Conversely, no seafood species examined exhibited significant risk from cadmium and mercury. These findings are consistent with the work of Arampongpun [33], who reported that striped snakehead fish available in Bangkok markets was free from cadmium-related health hazards. Similarly, another prior study demonstrated low levels of risk from cadmium and mercury in both freshwater and marine fish in Thailand, while highlighting the potential health risk associated with arsenic exposure from striped snakehead fish [10]. Nearly all of the seafood species examined in this study presented a low level of risk associated with arsenic exposure, while none demonstrated risk from cadmium and mercury. For instance, Northern whiting fish and Silver pomfret were determined to be safe for human consumption, indicating negligible concern regarding adverse effects from these toxic elements. These findings are in agreement with the study of Juwa [35], which similarly reported that walking catfish from Kwan-Phayao posed no significant health risk from cadmium exposure [33].

Only in conditions with significant arsenic concentrations did wedge shell and Musk crabs show signs of a possible arsenic danger. While the highest quantity of arsenic found in other seafood species was 0.17 mg/kg, the 97.5th percentile of arsenic content in these species was 0.25 mg/kg, which is similar to the amount found in red tilapia (0.05–0.25 mg/kg). All species showed no danger from cadmium and mercury, which is in line with research by Dokmaikaw and Suntaravitun [36], who also found no negative health consequences linked to these heavy metals in red tilapia from the Chachoengsao municipal market. Seafood was identified as presenting a considerable health risk from arsenic exposure across all consumer groups, while no risk was observed from cadmium. With respect to mercury, a potential health risk was evident only when seafood contained elevated mercury concentrations. The study found that the largest amount of arsenic found in other species was 0.15 mg/kg, while the 97.5th percentile of seafood’s arsenic concentration was 0.53 mg/kg, mostly linked to wedge shell (0.22–0.55 mg/kg). These findings suggest that Wedge shell may pose a health concern due to arsenic exposure. The results are consistent with those of Thongra-ar [37], who reported that seafood from the coastal region of the Map Ta Phut industrial estate carried risks from Hg exposure but remained free of cadmium- and lead-related health hazards.

The wedge shell was linked to a significant risk of arsenic exposure alone in instances of increased consumption, but no danger was detected from cadmium, mercury, or lead. These results are in line with those of Arbsuwan [38], who found no health hazards associated with cadmium in some marine species, such as Indo-Pacific mackerel that was caught at the pier in the Khlong Yai district. Ritonga et al. [39] also showed that Indian mackerel, which belongs to the same family as Indo-Pacific mackerel and was tested from Bangkok markets, was protected from harmful health effects caused by mercury. The majority of seafood samples demonstrated a high risk of concern with respect to arsenic exposure. When analyzed by age group, children aged 0–2.9 years and 3–5.9 years exhibited lower margins of exposure compared with older groups. For instance, the consumption of shellfish and crab posed a particularly high risk among those aged 0–2.9 years, underscoring the importance of evaluating cumulative risk across all seafood categories. To ensure accuracy, the margin of exposure was calculated for each age group considering the intake of all seafood species, as consumers typically consume a variety of seafood rather than a single type. The findings revealed that the margin of exposure for children aged 0–2.9 years and 3–5.9 years was below 100, indicating that arsenic exposure in this age range presents a substantial health risk warranting special attention. Nevertheless, further investigation into bioaccessibility, particularly of arsenic, is necessary to determine the extent to which arsenic can be released from the food matrix under human digestive conditions [40].

## 5. Conclusions

All analyzed toxic elements (As, Cd, Hg, and Pb) in the examined seafood were within the legal Thai standards, with the exception of Wedge shell and Musk crab, which contained arsenic levels exceeding the permissible limit. Risk assessment indicated no health risks associated with cadmium across the studied species and minimal concern for cadmium except in Wedge shell. For lead, based on the margin of exposure (*MOE*) approach, it indicated health risks for eater only group. The *MOE* values for lead in all seafood species for eater only group were below the threshold of concern (*MOE* < 100), confirming that lead exposure through seafood consumption poses significant health risk for eater only group. Wedge shell and Musk crab presented a high level of concern with respect to arsenic exposure. Shrimp and prawn were identified as high-risk species primarily among individuals with elevated consumption levels, whereas the risk from Wedge shell and Musk crab was evident only when these species contained high arsenic concentrations. Notably, children aged 0–5.9 years were recognized as a particularly vulnerable group, demonstrating a high risk of concern from arsenic when consuming Wedge shell and Musk crab. These findings suggest potential adverse health effects from toxic element exposure through seafood consumption. In addition, certain seafood species posed a high risk from mercury exposure, particularly among high-consuming groups when mercury levels were elevated. Therefore, providing consumers with information regarding foods potentially contaminated with toxic elements is essential to mitigate health risks. Furthermore, routine monitoring of toxic element contamination in high-risk seafood products should be carried out by food safety authorities.

## Figures and Tables

**Table 1 foods-14-03725-t001:** List of the 20 seafood species used in the study.

Common Name	Scientific Name	Local Name	Purchase
			(Month/Year)
Shrimp and prawn (captured)			
Pacific white shrimp	*Litopenaeus vannamei*	Koong Khaw	2/2025
Banana prawn	*Fenneropenaeus merguiensis*	Koong Share Buay	2/2025
Giant Tiger Prawn	*Penaeus monodon*	Koong Kula Dam	2/2025
Ornate rock lobster	*Panulirus ornatus*	Koong Mungkorn	2/2025
Crabs (captured)			
Musk Crab	*Charybdis feriata Linnaeus*	Pu Lai Sua	2/2025
Blue crab	*Portunus pelagicus*	Pu Ma	3/2025
Serrated Mud Crab	*Scylla serrata*	Pu Dam	3/2025
Red frog crab	*Ranina vanima*	Pu Juckajun	3/2025
Squids (captured)			
Splendid squid	*Loligo duvauceli*	Pla Muek Kluay	3/2025
Cuttlefish	*Sepia brevimana*	Pla Muek Kradong	3/2025
Bigfin reef squid	*Sepioteuthis lessoniana*	Pla Muek Hom	3/2025
Shellfish (captured)			
Razor clam	*Solen strictus Gould*	Hoi Lord	3/2025
Oysters	*Crassostrea gigas*	Hoi Nang Rom	3/2025
Cockle	*Tegillarca granosa*	Hoi Krang	3/2025
Clam	*Paphia undulata*	Hoi Lai	3/2025
Mussels	*Perna viridis*	Hoi Malang Poo	3/2025
Wedge shell	*Mercenaria mercenaria*	Hoi Talab	3/2025
Indo-Pacific horseshoe crab (eggs)	*Tachypleus gigas*	Mangda Jan	3/2025
Marine fish (farmed)			
Northern whiting fish	*Sillago sihama*	Pla Hed Kone	3/2025
Silver pomfret	*Pampus argenteus*	Pla Jaramed Khaw	3/2025

**Table 2 foods-14-03725-t002:** Health-based guidance value of heavy metals.

Heavy Metals	*HBGV*s	Toxicological Endpoint
Arsenic	*BMDL*	0.06 µg/kg BW/day (skin, bladder, lung cancer)
Cadmium	*PTMI*	25 µg/kg BW/monthly (renal tubular dysfunction)
Mercury	*PTWI*	4 µg/kg BW/weekly (renal toxicity)
Lead	*BMDL*	0.5 µg/kg BW/day (3–17.9 y developmental neurotoxicity) 0.63 µg/kg BW/day (≥18 y chronic kidney disease)

**Table 3 foods-14-03725-t003:** Consumption data of seafood (per capita).

Sample	Food Consumption	Age Group (Year Old) (g/Person/Day)					
		3 to 5.9	6 to 12.9	13 to 17.9	18 to 34.9	35 to 64.9	65 and Older
Shrimp and prawn	average	3.95	5.2	5.14	4.55	2.65	1.5
	97.5th percentile	27.43	41.15	27.43	27.43	20.57	13.71
Crabs	average	0.7	0.82	0.81	1.19	0.86	0.27
	97.5th percentile	6.29	8.8	6.29	9.43	6.29	2.93
Squids	average	2.8	4.61	5.31	5.42	2.36	0.89
	97.5th percentile	20.57	27.43	41.14	34.29	13.72	6.86
Razor clam	average	0.04	0.16	0.06	0.07	0.06	0.04
	97.5th percentile	0.00	0.47	0.00	0.00	0.00	0.00
Oysters	average	0.02	0.08	0.52	0.63	0.2	0.11
	97.5th percentile	0.00	0.00	4.40	6.60	2.20	0.00
Cockle	average	0.97	1.71	1.82	2.06	1.02	0.32
	97.5th percentile	5.14	20.57	20.57	20.57	10.29	2.57
Clam	average	0.26	0.51	0.52	0.75	0.62	0.24
	97.5th percentile	2.93	5.86	5.86	8.2	5.86	2.73
Mussels	average	0.59	0.92	0.83	1.46	0.79	0.27
	97.5th percentile	8.00	8.00	8.00	12.00	8.00	2.80
Wedge shell	average	0.04	0.16	0.06	0.07	0.06	0.04
	97.5th percentile	0.00	0.47	0.00	0.00	0.00	0.00
Indo-Pacific horseshoe crab (eggs)	average	0.06	0.24	0.13	0.16	0.12	0.11
	97.5th percentile	0.73	3.14	1.47	1.57	1.47	1.47
Marine fish	average	0.69	0.79	0.71	1.7	1.45	0.88
	97.5th percentile	7.14	7.14	10.29	24	14.29	7.14

**Table 4 foods-14-03725-t004:** Consumption data of seafood (Eater-only).

Sample	Food Consumption	Age Group (Year Old) (g/Person/Day)					
		3 to 5.9	6 to 12.9	13 to 17.9	18 to 34.9	35 to 64.9	65 and Older
Shrimp and prawn	average	44.63	59.98	55.16	59.13	49.04	40.01
	97.5th percentile	96.00	96.00	96.00	96.00	96.00	96.00
Crabs	average	30.13	37.27	41.9	50.59	52.66	36.55
	97.5th percentile	44.00	88.00	132.00	176.00	154.00	88.00
Squids	average	31.47	43.14	43.05	47.39	39.81	29.27
	97.5th percentile	80.00	96.00	96.00	96.00	96.00	80.00
Razor clam	average	27.75	26.76	28.21	40.1	29.18	20.71
	97.5th percentile	56.00	56.00	56.00	84.00	56.00	28.00
Oysters	average	34.13	57.13	65.46	66.58	56.26	43.95
	97.5th percentile	132.00	66.00	132.00	132.00	132.00	66.00
Cockle	average	33.56	44.16	46.15	49.76	47.19	35.42
	97.5th percentile	72.00	144.00	72.00	144.00	108.00	72.00
Clam	average	32.64	41.62	47.14	50.68	43.19	32.64
	97.5th percentile	82.00	82.00	123.00	82.00	82.00	61.50
Mussels	average	23.58	32.03	34.49	38.35	36.28	27.92
	97.5th percentile	56.00	56.00	56.00	112.00	140.00	56.00
Wedge shell	average	27.75	26.76	28.21	40.1	29.18	20.71
	97.5th percentile	56.00	56.00	56.00	84.00	56.00	28.00
Indo-Pacific horseshoe crab (eggs)	average	10.67	14.38	14.58	15.05	12.85	12.56
	97.5th percentile	22.00	44.00	44.00	44.00	22.00	44.00
Marine fish	average	29.55	42.18	42.74	46.95	41.25	33.74
	97.5th percentile	50.00	100.00	100.00	100.00	100.00	96.00

**Table 5 foods-14-03725-t005:** Average consumer body weight by age group.

**Age group (Year Old)**	3 to 5.9	6 to 12.9	13 to 17.9	18 to 34.9	35 to 64.9	65 and older
**Body Weight (kg)**	17.25	33.38	53.42	63.12	63.53	55.77

**Table 6 foods-14-03725-t006:** Moisture content and concentrations of toxic elements for each seafood species. Data are reported as mean ^1^ ± SD (*n* = 3).

Seafood Species	Moisture (%)	Toxic Element Content (mg/kg Fresh Weight)			
		As	Cd	Hg	Pb
Pacific white shrimp	76.1 ± 0.0	1.7 ± 0.0 ^i^	0.1 ± 0.0 ^c^	ND ^2^	0.1 ± 0.1 ^a^
Banana prawn	81.5 ± 0.9	1.8 ± 0.3 ^h,i^	0.1 ± 0.0 ^c^	<LOQ ^3^	0.1 ± 0.1 ^a^
Giant Tiger Prawn	84.4 ± 0.6	1.5 ± 0.2 ^i,j^	0.1 ± 0.0 ^c^	ND ^2^	0.1 ± 0.1 ^a^
Ornate rock lobster	79.4 ± 0.3	9.0 ± 0.6 ^d^	0.3 ± 0.6 ^c^	0.1 ± 0.0 ^a^	0.1 ± 0.1 ^a^
Musk Crab	82.7 ± 0.1	16.2 ± 1.8 ^b^	1.1 ± 0.7 ^b^	0.1 ± 0.0 ^a^	0.1 ± 0.0 ^a^
Blue crab	81.3 ± 0.5	2.8 ± 0.2 ^g^	0.4 ± 0.1 ^c^	0.1 ± 0.0 ^a^	0.1 ± 0.0 ^a^
Serrated Mud Crab	75.2 ± 0.3	2.4 ± 0.8 ^g^	0.3 ± 0.5 ^c^	0.1 ± 0.0 ^a^	0.1 ± 0.0 ^a^
Red frog crab	83.3 ± 0.3	0.2 ± 0.3 ^k^	0.1 ± 0.0 ^c^	ND ^2^	0.1 ± 0.0 ^a^
Splendid squid	82.7 ± 0.9	1.7 ± 0.2 ^i^	0.8 ± 0.4 ^b,c^	0.1 ± 0.0 ^a^	0.2 ± 0.0 ^a^
Cuttlefish	83.5 ± 0.2	3.4 ± 0.0 ^f^	0.1 ± 0.0 ^c^	0.1 ± 0.2 ^a^	0.1 ± 0.1 ^a^
Bigfin reef squid	87.2 ± 0.7	3.0 ± 0.7 ^f^	0.3 ± 0.0 ^c^	0.1 ± 0.0 ^a^	0.1 ± 0.0 ^a^
Razor clam	78.6 ± 0.2	6.9 ± 0.8 ^e^	0.1 ± 0.0 ^c^	0.1 ± 0.2 ^a^	0.1 ± 0.0 ^a^
Oysters	78.6 ± 0.7	2.0 ± 0.6 ^g,h^	0.2 ± 0.8 ^c^	0.1 ± 0.2 ^a^	0.1 ± 0.1 ^a^
Cockle	88.0 ± 0.0	1.3 ± 0.6 ^j^	1.7 ± 0.3 ^b^	0.1 ± 0.0 ^a^	0.2 ± 0.0 ^a^
Clam	81.7 ± 0.2	1.5 ± 0.2 ^i,j^	1.3 ± 0.6 ^b^	0.1 ± 0.0 ^a^	0.1 ± 0.0 ^a^
Mussels	86.5 ± 0.1	1.4 ± 0.3 ^i,j^	0.3 ± 0.2	0.1 ± 0.0 ^a^	0.1 ± 0.1 ^a^
Wedge shell	77.0 ± 0.4	21.4 ± 2.0 ^a^	7.3 ± 1.8 ^a^	0.1 ± 0.1 ^a^	0.1 ± 0.0 ^a^
Indo-Pacific horseshoe crab (eggs)	64.1 ± 0.2	10.6 ± 1.7 ^c^	0.9 ± 0.5 ^b,c^	0.1 ± 0.0 ^a^	0.1 ± 0.0 ^a^
Northern whiting fish	75.2 ± 0.8	3.8 ± 0.0 ^f^	0.1 ± 0.0 ^c^	0.1 ± 0.0 ^a^	0.1 ± 0.0 ^a^
Silver pomfret	80.0 ± 0.0	0.7 ± 0.4 ^i^	0.1 ± 0.0 ^c^	ND ^2^	0.1 ± 0.0 ^a^

^1^ presented as mean ± SD from 3 market; ^2^ ND indicates values not detected (below the limit of detection, LOD, or < 0.001 mg/kg); ^3^ <LOQ denotes concentrations below the limit of quantification—specifically, <0.006 mg/kg for arsenic and cadmium, and <0.012 mg/kg for mercury and lead. Values in the same column bearing different superscript letters represent statistically significant differences among seafood species for the given variable (*p* < 0.05), as determined by one-way ANOVA followed by Duncan’s multiple range post hoc test. Maximum Residue Limits (MRLs) by CODEX for As, Cd, Pb, and Hg is <0.1mg/kg, <0.05–0.1 mg/kg, <0.3 mg/kg, and Hg < 0.5 mg/kg, respectively.

**Table 7 foods-14-03725-t007:** (**a**) Each seafood species’ arsenic exposure margin (per capita). (**b**) Margin of exposure of arsenic from each seafood species (Per capita). (**c**) Margin of exposure of arsenic from each seafood species (Per capita). (**d**) Margin of exposure of arsenic from each seafood species (Per capita).

(**a**)									
**Type of Sample**		**Cooking Method**	**Food Consumption**	**Age Group (Year Old)**					
				**3 to 5.9**	**6 to 12.9**	**13 to 17.9**	**18 to 34.9**	**35 to 64.9**	**65 and Older**
Shrimp and prawn (captured)	Pacific white shrimp	Boiled	average	1.8	2.7	4.4	5.9	10.1	15.7
			97.5th percentile	0.3	0.3	0.8	1.0	1.3	1.7
		Fried	average	2.2	3.2	5.1	6.8	11.8	18.3
			97.5th percentile	0.3	0.4	1.0	1.1	1.5	2.0
		Grilled	average	2.2	3.2	5.1	6.8	11.8	18.3
			97.5th percentile	0.3	0.4	1.0	1.1	1.5	2.0
	Banana prawn	Boiled	average	1.7	2.5	4.1	5.5	9.4	14.6
			97.5th percentile	0.2	0.3	0.8	0.9	1.2	1.6
		Fried	average	2.0	2.9	4.8	6.4	11.0	17.1
			97.5th percentile	0.3	0.4	0.9	1.1	1.4	1.9
		Grilled	average	2.0	2.9	4.8	6.4	11.0	17.1
			97.5th percentile	0.3	0.4	0.9	1.1	1.4	1.9
	Giant Tiger Prawn	Boiled	average	2.4	3.5	5.7	7.7	13.2	20.5
			97.5th percentile	0.3	0.4	1.1	1.3	1.7	2.2
		Fried	average	2.1	3.0	4.9	6.6	11.3	17.6
			97.5th percentile	0.3	0.4	0.9	1.1	1.5	1.9
		Grilled	average	2.1	3.0	4.9	6.6	11.3	17.6
			97.5th percentile	0.3	0.4	0.9	1.1	1.5	1.9
	Ornate rock lobster	Boiled	average	0.3	0.5	0.8	1.1	1.9	3.0
			97.5th percentile	0.05	0.06	0.2	0.2	0.2	0.3
		Fried	average	0.5	0.7	1.1	1.5	2.5	3.9
			97.5th percentile	0.1	0.1	0.2	0.2	0.3	0.4
		Grilled	average	0.3	0.4	0.7	0.9	1.6	2.5
			97.5th percentile	0.04	0.1	0.1	0.2	0.2	0.3
Crabs (captured)	Musk Crab	Boiled	average	1.1	1.8	2.9	2.3	3.3	9.1
			97.5th percentile	0.1	0.2	0.4	0.3	0.4	0.8
		Fried	average	0.7	1.2	1.9	1.6	2.2	6.1
			97.5th percentile	0.1	0.1	0.3	0.2	0.3	0.6
		Grilled	average	0.9	1.5	2.4	2.0	2.7	7.6
			97.5th percentile	0.1	0.1	0.3	0.2	0.4	0.7
(**b**)									
**Type of Sample**		**Cooking Method**	**Food Consumption**	**Age Group (Year Old)**					
				**3 to 5.9**	**6 to 12.9**	**13 to 17.9**	**18 to 34.9**	**35 to 64.9**	**65 and Older**
Crabs (captured)	Blue crab	Boiled	average	7.3	12.1	19.6	15.8	22.0	61.5
			97.5th percentile	0.8	1.1	2.5	2.0	3.0	5.7
		Fried	average	5.2	8.7	14.0	11.3	15.7	43.9
			97.5th percentile	0.6	0.8	1.8	1.4	2.1	4.0
		Grilled	average	5.2	8.7	14.0	11.3	15.7	43.9
			97.5th percentile	0.6	0.8	1.8	1.4	2.1	4.0
	Serrated Mud Crab	Boiled	average	8.3	13.8	22.3	18.0	25.0	69.9
			97.5th percentile	0.9	1.3	2.9	2.3	3.4	6.4
		Fried	average	8.3	13.8	22.3	18.0	25.0	69.9
			97.5th percentile	0.9	1.3	2.9	2.3	3.4	6.4
		Grilled	average	8.3	13.8	22.3	18.0	25.0	69.9
			97.5th percentile	0.9	1.3	2.9	2.3	3.4	6.4
	Red frog crab	Boiled	average	12.0	19.9	32.2	25.9	36.1	100.8
			97.5th percentile	1.3	1.9	4.1	3.3	4.9	9.3
		Fried	average	10.3	17.0	27.6	22.2	30.9	86.4
			97.5th percentile	1.1	1.6	3.6	2.8	4.2	8.0
		Grilled	average	12.0	19.9	32.2	25.9	36.1	100.8
			97.5th percentile	1.3	1.9	4.1	3.3	4.9	9.3
Squids (captured)	Splendid squid	Boiled	average	1.5	1.8	2.5	2.9	6.6	15.5
			97.5th percentile	0.2	0.3	0.3	0.5	1.1	2.0
		Fried	average	1.1	1.3	1.8	2.1	4.7	11.0
			97.5th percentile	0.1	0.2	0.2	0.3	0.8	1.4
		Grilled	average	1.7	2.0	2.8	3.3	7.6	17.7
			97.5th percentile	0.2	0.3	0.4	0.5	1.3	2.3
	Cuttlefish	Boiled	average	1.7	2.0	2.8	3.2	7.4	17.1
			97.5th percentile	0.2	0.3	0.4	0.5	1.3	2.2
		Fried	average	1.2	1.4	2.0	2.3	5.3	12.2
			97.5th percentile	0.2	0.2	0.3	0.4	0.9	1.6
		Grilled	average	1.7	2.0	2.8	3.2	7.4	17.1
			97.5th percentile	0.2	0.3	0.4	0.5	1.3	2.2
(**c**)									
**Type of Sample**		**Cooking Method**	**Food Consumption**	**Age Group (Year Old)**					
				**3 to 5.9**	**6 to 12.9**	**13 to 17.9**	**18 to 34.9**	**35 to 64.9**	**65 and Older**
Squids (captured)	Bigfin reef squid	Boiled	average	0.7	0.9	1.2	1.4	3.2	7.5
			97.5th percentile	0.1	0.1	0.2	0.2	0.6	1.0
		Fried	average	0.6	0.7	1.0	1.2	2.8	6.5
			97.5th percentile	0.1	0.1	0.1	0.2	0.5	0.8
		Grilled	average	0.6	0.7	1.0	1.2	2.8	6.5
			97.5th percentile	0.1	0.1	0.1	0.2	0.5	0.8
Shellfish (captured)	Razor clam	Boiled	average	200.2	96.9	413.3	418.6	491.6	647.3
			97.5th percentile	-	33.0	-	-	-	-
		Fried	average	175.2	84.7	361.7	366.3	430.1	566.4
			97.5th percentile	-	28.8	-	-	-	-
		Grilled	average	175.2	84.7	361.7	366.3	430.1	566.4
			97.5th percentile	-	28.8	-	-	-	-
	Oysters	Boiled	average	379.6	183.6	45.2	44.1	139.8	223.2
			97.5th percentile	-	-	5.3	4.2	12.7	-
		Fried	average	303.7	146.9	36.2	35.3	111.8	178.5
			97.5th percentile	-	-	4.3	3.4	10.2	-
		Grilled	average	227.8	110.2	27.1	26.5	83.9	133.9
			97.5th percentile	-	-	3.2	2.5	7.6	-
	Cockle	Boiled	average	7.0	7.7	11.5	12.0	24.4	68.4
			97.5th percentile	1.3	0.6	1.0	1.2	2.4	8.5
		Fried	average	5.6	6.1	9.2	9.6	19.5	54.7
			97.5th percentile	1.1	0.5	0.8	1.0	1.9	6.8
		Grilled	average	2.8	3.1	4.6	4.8	9.8	27.4
			97.5th percentile	0.5	0.3	0.4	0.5	1.0	3.4
	Clam	Boiled	average	207.1	204.3	320.6	262.7	319.8	725.2
			97.5th percentile	18.4	17.8	28.5	24.0	33.8	63.8
		Fried	average	138.0	136.2	213.7	175.1	213.2	483.5
			97.5th percentile	12.2	11.9	19.0	16.0	22.6	42.5
		Grilled	average	172.6	170.2	267.2	218.9	266.5	604.4
			97.5th percentile	15.3	14.8	23.7	20.0	28.2	53.1
(**d**)									
**Type of Sample**		**Cooking Method**	**Food Consumption**	**Age Group (Year Old)**					
				**3 to 5.9**	**6 to 12.9**	**13 to 17.9**	**18 to 34.9**	**35 to 64.9**	**65 and Older**
Shellfish (captured)	Mussels	Boiled	average	12.2	15.1	26.9	18.0	33.5	86.2
			97.5th percentile	0.9	1.7	2.8	2.2	3.3	8.3
		Fried	average	7.3	9.1	16.1	10.8	20.1	51.7
			97.5th percentile	**0.5**	1.0	1.7	1.3	2.0	5.0
		Grilled	average	7.3	9.1	16.1	10.8	20.1	51.7
			97.5th percentile	**0.5**	1.0	1.7	1.3	2.0	5.0
	Wedge shell	Boiled	average	14.5	7.0	30.0	30.3	35.6	46.9
			97.5th percentile	-	2.4	-	-	-	-
		Fried	average	12.1	5.8	25.0	25.3	29.7	39.1
			97.5th percentile	-	2.0	-	-	-	-
		Grilled	average	14.5	7.0	30.0	30.3	35.6	46.9
			97.5th percentile	-	2.4	-	-	-	-
	Indo-Pacific horseshoe crab (eggs)	Boiled	average	29.1	14.1	41.6	39.9	53.6	51.3
			97.5th percentile	2.4	1.1	3.7	4.1	4.4	3.8
		Fried	average	22.6	10.9	32.3	31.1	41.7	39.9
			97.5th percentile	1.9	0.8	2.9	3.2	3.4	3.0
		Grilled	average	25.9	12.5	37.0	35.5	47.6	45.6
			97.5th percentile	2.1	1.0	3.3	3.6	3.9	3.4
Marine fish (farmed)	Northern whiting fish	Boiled	average	17.7	30.0	53.3	26.3	31.1	44.9
			97.5th percentile	1.7	3.3	3.7	1.9	3.2	5.5
		Fried	average	9.8	16.6	29.6	14.6	17.3	25.0
			97.5th percentile	1.0	1.8	2.0	1.0	1.8	3.1
		Grilled	average	11.8	20.0	35.6	17.5	20.7	29.9
			97.5th percentile	1.1	2.2	2.5	1.2	2.1	3.7
	Silver pomfret	Boiled	average	90.3	152.6	271.8	134.1	158.3	228.9
			97.5th percentile	8.7	16.9	18.8	9.5	16.1	28.2
		Fried	average	50.2	84.8	151.0	74.5	87.9	127.2
			97.5th percentile	4.8	9.4	10.4	5.3	8.9	15.7
		Grilled	average	70.2	118.7	211.4	104.3	123.1	178.1
			97.5th percentile	6.8	13.1	14.6	7.4	12.5	21.9

## Data Availability

The original contributions presented in this study are included in the article and Appendix A. Further inquiries can be directed to the corresponding author.

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
