# Peer review of "Risk Assessment of Toxic Heavy Metal Exposure in Selected Seafood Species from Thailand"

_foods, 2025, doi:10.3390/foods14213725_

Round 1

Reviewer 1 Report

Comments and Suggestions for Authors

This article deals with the main topic of the evaluation of the concentration of some potentially toxic heavy metals in different seafood products associated with a risk assessment carried out on different categories of people. The topic is not particularly innovative since there are already numerous publications that follow this classic scheme, furthermore, the same authors published an identical work in Foods in 2023, modifying only the species analyzed. However, it is important to periodically carry out monitoring studies to understand the environmental situation of certain regions of a country and the related risk for the population.

Introduction

The introduction is sufficiently detailed and reports an appropriate state of the art.

Materials and Methods

In line 110, it is reported that the samples were freeze-dried before being analyzed. However, in line 158, the authors report that the data were expressed as fresh weight (FW). The reason for this is unclear, and above all, the authors need to explain the calculations used to convert from dry weight (DW) to FW.

Line 159: The authors performed recovery tests that were fairly insignificant and of little scientific value, as the samples were completely mineralized by acid digestion. Recoveries make sense in extraction processes.

Line 205: Why was body weight not taken into account in equation 4 as in equation 1?

Line 197 and table 3: In line 197 it is reported that for MOE values ​​above 100 there are no risks, however the reasoning behind why in table 3 there are values ​​in bold, while other values ​​below 100 are not is not clear.

Line 190 states: "The HBGV for mercury corresponds to a tolerable weekly intake of 25 µg/kg body weight," however, line 363 states: "The risk assessment of mercury using the Provisional Tolerable Weekly Intake (PTWI), which in this case is 4 µg/kg body weight per week." The authors should clarify this point.

Results

In Table 2, a column should be added with the MRLs for the various metals (if present)

Readers should be able to independently perform risk assessment calculations. Authors should report (perhaps in supplemental tables) daily intakes for different seafood products, body weights for different population groups, and metal concentrations in commodities after cooking.

In particular, these last data would be interesting to see if there is any variation between the raw and cooked samples, perhaps adding a figure with histograms and a statistical treatment.

In addition, the reviewer would like to access the raw data to verify how the risk assessment calculations were performed.

Residue data and risk assessment for the other three metals (cadmium, lead and mercury) are completely missing.

Discussion

For a smoother reading, remove the paragraphs and leave the discussion in its entirety

Line 416 - 429. Delete because this part is a repetition of information already reported above.

Lines 443 and 441. Delete repetitions

Given the vastness of the data, reading the discussion would be made easier by including a table summarizing the bibliographic references with the species studied by other authors and the concentrations of heavy metals found both in Thailand and in other countries.

Author Response

  1. The introduction is sufficiently detailed and reports an appropriate state of the art.

Response: Thank you very much for all comments and suggestion.

  1. In line 110, it is reported that the samples were freeze-dried before being analyzed. However, in line 158, the authors report that the data were expressed as fresh weight (FW). The reason for this is unclear, and above all, the authors need to explain the calculations used to convert from dry weight (DW) to FW.

Response: For mineral analysis, homogenized samples were weighed at less than 0.5 g freeze-dried prior to microwave digestion. The weights of the samples before and after freeze-drying were recorded. Toxic element concentrations were determined by ICP-MS and reported in units of µg/kg based on the dried sample. To express the results on a fresh weight (FW) basis, concentrations were converted using the ratio of freeze-dry weight (DW) to fresh weight.
For example, if the Cd concentration measured by ICP-MS is 800 µg/kg in the freeze-dried sample, with the fresh sample weight of 400 g and the dry sample weight of 100 g, the Cd concentration on a fresh weight basis is calculated as follows:
           CdFW = 800×100/400=200 μg/kg

   which corresponds to 20 µg per 100 g FW.

More information was added in the section 2.1 Seafood Sample Purchasing and Preparation.

  1. Line 159: The authors performed recovery tests that were fairly insignificant and of little scientific value, as the samples were completely mineralized by acid digestion. Recoveries make sense in extraction processes.

Response: The % recovery values obtained in this study were used to demonstrate the accuracy of the analytical method, confirming that there was no significant loss of the target elements during sample digestion, dilution, and quantification processes. This ensures the reliability of the measurements.

  1. Line 205: Why was body weight not taken into account in equation 4 as in equation 1?

Response: Thank you for highlighting this point. For lead, Equation 4 followed the FDA Interim Reference Level in micrograms per day, so body weight normalization was not applied. To harmonize across elements, we now also report a body weight normalized margin of exposure for Pb, analogous to iAs, and we have clarified this in the Methods and revised Section 3.3.5  Risk Assessment of Pb accordingly.

  1. Line 197 and table 3: In line 197 it is reported that for MOE values ​​above 100 there are no risks, however the reasoning behind why in table 3 there are values ​​in bold, while other values ​​below 100 are not is not clear.

Response: Thank you for pointing out this inconsistency. The statement that MOE values above 100 indicate no risk applies to our Pb context, whereas for inorganic arsenic we follow the updated EFSA assessment of January 18, 2024, which defines much lower MOE thresholds for concern. We have revised the text in Lines 270–273 and Section 2.5  to make this explicit, and we have adjusted Table 3 so that boldface appears only for values that indicate potential concern under the appropriate criterion for each element. Citations to the EFSA update have been added for transparency.

  1. Line 190 states: "The HBGV for mercury corresponds to a tolerable weekly intake of 25 µg/kg body weight," however, line 363 states: "The risk assessment of mercury using the Provisional Tolerable Weekly Intake (PTWI), which in this case is 4 µg/kg body weight per week." The authors should clarify this point.

Response: Thank you for catching this discrepancy. The correct value for mercury is the PTWI of 4 µg/kg body weight per week, and the 25 µg/kg body weight figure pertains to cadmium. The mention of 25 for mercury in Line 190  was an error. We have corrected the manuscript throughout, clarified this in Section 2.5, and added the appropriate references.

  1. In Table 2, a column should be added with the MRLs for the various metals (if present)

Response: Table 2. Health based guidance value of heavy metals was revised.

  1. Readers should be able to independently perform risk assessment calculations. Authors should report (perhaps in supplemental tables) daily intakes for different seafood products, body weights for different population groups, and metal concentrations in commodities after cooking.

Response: Thank you for this helpful suggestion. We have added the requested inputs in Section 2.5, specifically Table 3, consumption data for seafood per capita, Table 4, consumption data for seafood among eater only consumers, and Table 5, average body weight by age group. To obtain metal concentrations in commodities after cooking, we applied documented cooking factors to the measured values, and we now describe the DW to FW and cooking factor conversions in Section 2.5 with full citations. We note that readers in other settings may substitute their local values.

  1. In particular, these last data would be interesting to see if there is any variation between the raw and cooked samples, perhaps adding a figure with histograms and a statistical treatment.

Response: The effect of different cooking methods on heavy metal retention is a limitation of this study, and it should be addressed in future research.

  1. In addition, the reviewer would like to access the raw data to verify how the risk assessment calculations were performed.

Response: Thank you for this request. We have provided the raw dataset in Excel to the editors and reviewers to enable full verification of the calculations.

  1. Residue data and risk assessment for the other three metals (cadmium, lead and mercury) are completely missing.

Response: These results are not missing. To keep the manuscript concise, we deliberately present full risk tables only for inorganic arsenic as an exemplar, because including all metals and cooking methods would produce an unwieldy number of tables. For cadmium, lead, and mercury, the risk assessment is described in the text rather than shown as tables.

  1. For a smoother reading, remove the paragraphs and leave the discussion in its entirety

Response: The text was amended as suggested.

  1. Line 416 - 429. Delete because this part is a repetition of information already reported above.

Response: “Risk characterization employed the MOE approach, utilizing a BMDL value of 0.06 µg/kg BW/day established by EFSA. According to EFSA, MOE thresholds indicating health concerns range from 2.0 to 0.4 for average consumers and 0.9 to 0.2 for high consumers, suggesting potential health risks that require careful monitoring [23].” was deleted.

  1. Lines 443 and 441. Delete repetitions

Response: May we kept these sentences to emphasize this point.

  1. Given the vastness of the data, reading the discussion would be made easier by including a table summarizing the bibliographic references with the species studied by other authors and the concentrations of heavy metals found both in Thailand and in other countries.

Response: We appreciate the reviewer’s suggestion to include a summary table of bibliographic references and heavy metal concentrations from other studies. However, the Discussion section already provides a comprehensive comparison of the results with previous studies conducted both in Thailand and abroad. Adding another summary table might result in redundancy and make the section excessively long. Therefore, we have retained the comparative information in the text to maintain readability and logical flow.

Reviewer 2 Report

Comments and Suggestions for Authors

The manuscript provides information on the heavy metal content and health risks of consuming some seafood cooked in three ways, finding that Hg is the most prevalent ion. This information is important for local consumers, although the authors do not indicate whether the products are exported, which could have a greater impact. Regarding the organization of the manuscript, it follows the journal's standards. The writing is clear.

However, there are some aspects that should be taken into account before final publication.

  1. L71-73. The authors could also provide information regarding other regulations, such as the WHO, among others.
  2. What were the digestion conditions in the digester? Also, for L156, the operating conditions.
  3. EN 2.4 refers to total or dissolved ions; please indicate this clearly.
  4. Why do these species have higher levels of Hg? Is this associated with their diet, habitat, or some other source? Although the products have been purchased, this higher content must be justified. This is explained in Section 4.1; however, I suggest expanding on this point clearly.
  5. The metal content in palm oil must be shown, and its contribution (if applicable) must be explained.

Author Response

Please see attached file for all comments and suggestion of each reviewer.

Reviewer 3 Report

Comments and Suggestions for Authors

1. The author should explain the rationale for selecting these elements.
2. The authors should specify the number of samples collected and the quantity per category.
3. ICP-MS detection parameters should be presented in tabular form. Spiked samples for method recovery assessment should include low, medium, and high concentration levels.
4. Elemental HBGVs should be tabulated, with toxicological endpoints clearly defined.
5. Does the “C” in the declaration denote only the average concentration, specifically the mean concentration across all samples?
6. Data such as food consumption rates and body weights should be explicitly stated in the declaration.
7. What do “abc” denote in Table 2?
8. Age groups should be specified as 3–6 (excluding 6), 6–13 (excluding 13), etc. Groups such as 5.9 or 12.9 are unacceptable as they render statistical analysis impossible.

Author Response

(The authors gave the same response as above.)

Round 2

Reviewer 1 Report

Comments and Suggestions for Authors

The manuscript, thanks to the timely contribution of the reviewers and the editor, has been significantly improved by the authors, however there are still some unclarified points.

Author response:

"The effect of different cooking methods on heavy metal retention is a limitation of this study, and it should be addressed in future research".

How was it possible to perform the risk assessment for the various commodities after the three firings if the residue data is not reported? I believe that for transparency, also as additional data, these input values ​​used in the risk assessment calculation should be reported. Only residues in the raw product were reported (new Table 6).

Author response:

"These results are not missing. To keep the manuscript concise, we deliberately present full risk tables only for inorganic arsenic as an exemplar, because including all metals and cooking methods would produce an unwieldy number of tables. For cadmium, lead, and mercury, the risk assessment is described in the text rather than shown as tables".

I understand the authors' need to present concise data in the manuscript; in fact, the supplementary data exist for that purpose. I believe that all data obtained during the study should be made available to readers in the form of tables.

Author response:

"The text was amended as suggested".

I had requested that the discussion be consolidated by removing the paragraph titles. The author replied that the correction had been made, but the discussion in the new manuscript is still divided into paragraphs.....

Author Response

Thank you for your review.

Reviewer 2 Report

Comments and Suggestions for Authors

Although most of the suggestions have been addressed, there are minor aspects that should be taken into account, such as indicating the operating conditions of the equipment used. I have no further comments.

Author Response

Thank you so much for your review.

Reviewer 3 Report

Comments and Suggestions for Authors

The author has made revisions to the manuscript based on the feedback. The quality of the manuscript has been significantly improved.

Author Response

Thank you so much for your review.